# Benchmarking tomographic acquisition schemes for high-resolution structural biology

Beata Turoňová[1], Wim J.H. Hagen[1], Martin Obr[2,6], Shyamal Mosalaganti[1], J. Wouter Beugelink[1,3,7], Christian E. Zimmerli[1,4], Hans-Georg Kräusslich[2] & Martin Beck[1,5✉]

Cryo electron tomography with subsequent subtomogram averaging is a powerful technique to structurally analyze macromolecular complexes in their native context. Although close to atomic resolution in principle can be obtained, it is not clear how individual experimental parameters contribute to the attainable resolution. Here, we have used immature HIV-1 lattice as a benchmarking sample to optimize the attainable resolution for subtomogram averaging. We systematically tested various experimental parameters such as the order of projections, different angular increments and the use of the Volta phase plate. We find that although any of the prominently used acquisition schemes is sufficient to obtain sub-nanometer resolution, dose-symmetric acquisition provides considerably better outcome. We discuss our findings in order to provide guidance for data acquisition. Our data is publicly available and might be used to further develop processing routines.

[1] European Molecular Biology Laboratory, Structure and Computational Biology Unit, 69124 Heidelberg, Germany. [2] Department of Infectious Diseases, Virology, Universitätsklinikum Heidelberg, 69120 Heidelberg, Germany. [3] Cryo-Electron Microscopy, Bijvoet Center for Biomolecular Research, Department of Chemistry, Faculty of Science, Utrecht University, 3584 CH Utrecht, The Netherlands. [4] Collaboration for joint PhD degree between EMBL and Heidelberg University, Faculty of Biosciences, 69124 Heidelberg, Germany. [5] Max Planck Institute of Biophysics, 60438 Frankfurt am Main, Germany. [6]Present address: Institute of Science and Technology Austria, A-3400 Klosterneuburg, Austria. [7]Present address: Crystal and Structural Chemistry, Bijvoet Center for Biomolecular Research, Department of Chemistry, Faculty of Science, Utrecht University, 3584 CH Utrecht, The Netherlands. ✉email: martin.beck@biophys.mpg.de

Cryo electron tomography (CryoET) is a powerful imaging technique to structurally analyze pleomorphic biological objects such as cells, organelles, and subcellular architecture[1,2]. In combination with subtomogram averaging (SA) structures of repetitive objects within such tomograms, such as e.g., macromolecular complexes, can be resolved[3,4]. In principle, close to atomic resolution can be obtained. In practice, however, although this technique is being used by many laboratories, the vast majority of structures are not resolved into the subnanometer regime. The biological properties of the object of interest are a prerequisite for obtaining high resolution. The most important of those properties are (i) specimen thickness, which is particularly critical for larger biological objects because it limits the attainable signal to noise ratio (SNR) at a given dose[5]. (ii) The abundance of the structure of interest within the pleomorphic objects that determines the number of repetitive subtomograms that can be obtained. (iii) The consistency of the structure across the repetitive objects, namely low structural dynamics. (iv) The structural preservation after embedding into vitrified ice[6]. (v) Lastly, the angular orientation of the given asymmetric unit should lead to isotropic sampling in Fourier space, otherwise the attainable resolution will be reduced along the respective spatial direction in real space.

Not only these biological properties but also technical parameters limit the attainable resolution. Unlike image acquisition for single particle analysis (SPA), tomographic data collection requires the specimen to be imaged at different tilt angles. This results in a number of complications that must be considered prior to the image acquisition. The total electron dose has to be distributed among the acquired projections leading to lower SNR when compared with SPA projections. The SNR decreases even more at high-tilt angles due to increased effective thickness of the sample. Moreover, the continued exposure results in an accumulation of dose and consequently the gradual deterioration of the specimen. As such, the information content decreases with the projection number whereby high-resolution information is lost at first[7]. In order to obtain the best possible resolution during the subsequent SA, one has to optimize the tilt range and the angular increment, thus defining the number of projections and the order in which they are acquired. Jointly, these parameters are referred to as a "tilt-scheme". Several previous studies have discussed how to choose the angular increment in order to obtain the best possible sampling of tomographic reconstructions in Fourier space[8,9]. The deductions from these studies are however not directly transferable to SA. In SA, the sampling of Fourier space is a result of averaging many subtomograms with different orientations within the tomograms of origin. Therefore, increasing the number of differently oriented subtomograms should be more important than uniform sampling of high-frequencies on the individual tomogram level.

Also for the order in which the projections are acquired, different tilt-schemes have been proposed. Traditionally, continuous acquisition schemes have been used. Here, the projections are collected by tilting strictly into one direction from a minimum tilt angle to the maximum tilt angle. The advantages of this scheme are the minimal mechanical interference during tilting and the relatively rapid data collection. However, the projections acquired at first and at the lowest accumulated dose, have a low SNR as they are collected at high tilts with large effective specimen thickness. One would predict that this caveat leads to a poor preservation of high-resolution information within the entire tomogram, although the impact of which has to the best of our knowledge not yet been systematically tested. To better deal with the trade-off of effective specimen thickness and accumulated dose, alternative schemes have been introduced. The bidirectional scheme[10] starts at 0°, or with an offset, and first proceeds toward

the minimum angle. Subsequently, it returns to the starting angle in order to continue to collect in positive direction until the maximum angle. This way the least dose-exposed projections are acquired where the effective specimen thickness is minimal, albeit in only one direction, which leads to better preservation of high-resolution information. The disadvantage of this approach is the difference in projection quality and resemblance between the first and the second half of the tilt-series, because the latter is only acquired after the specimen has already been exposed with half of the total dose. This can complicate the subsequent processing of the projections, especially in terms of tilt-series alignment[11]. To avoid any sharp decline of information content between adjacent projections, and in order to preserve as much high-resolution information as possible, the electron dose should be systematically accumulated from lower to higher tilt angles, and as such distributed symmetrically in both directions. The respective dose-symmetric (DS) tilt-scheme has been coined the "Hagen scheme"[12]. It starts the acquisition at 0° and then alternates positive and negative tilt angles until it reaches the specified range. In this way, the first projections containing the best-preserved high-resolution information, are acquired at low tilts and thus with the best possible SNR. In comparison to the aforementioned dose-asymmetric schemes, the DS scheme requires more acquisition time. How these different tilt-schemes affect the attainable resolution of SA has not yet been systematically tested.

Tilt-series are generally collected out of focus to generate phase contrast that facilitates particle detection but also leads to the signal modulation described by the contrast transfer function (CTF). CTF correction is required to properly interpret high-resolution structural features. The quality of the correction depends on the precision with which one is able to estimate the defocus for each projection. The high-tilt projections with rather low SNR are typically more difficult to correct, which is another argument for DS acquisition schemes. Alternatively, the Volta phase plate (VPP) allows contrast-rich imaging in focus without the need for CTF correction[13]. If a defocus is applied or observed because parts of the titled projections are above or below the focal plane, both defocus and phase-shift need to be determined prior to the CTF correction. Whether VPP projections are compatible with high-resolution SA has not yet been systematically tested.

As both biological properties of a sample and determination of optimal acquisition parameters play a key role in attainable resolution, it is difficult to assess in practice why structural analysis by SA is limited to a given resolution. Thus far, not many structures with subnanometer resolution were obtained by SA and only seven of those have reached a resolution below 5 Å (as of July 2019). The first one to breach the 5 Å barrier was a structure of the immature HIV-1 CA-SP1 lattice assembled in the presence of the maturation inhibitor Bevirimat (BVM), which was resolved to 3.9 Å[14]. The purified HIV-1 derived protein ΔMACANCSP2 forms virus-like particles (VLPs) in vitro, which exhibit an identical lattice as the immature HIV-1 capsid. These VLPs are well suited for SA. The specimen scores high on any of the five above-introduced biological parameters and thus represents an excellent object for the technical benchmarking of acquisition and processing routines. The particle has 120-nm diameter and is usually embedded into around 200-nm-thick ice. The VLPs contain a large copy number of the lattice-forming protein and the CA-SP1 layer of the protein forms a locally ordered shell with C6 symmetry. In the study reporting the 3.9 Å resolution, the DS scheme was used for the data collection[14], and it has been assumed that this scheme was critical for achieving the high resolution. Accordingly, it has been routinely used for samples with high-resolution potential and current all structures resolved below 5 Å were collected using this scheme. However, no systematic study/benchmarking was performed to

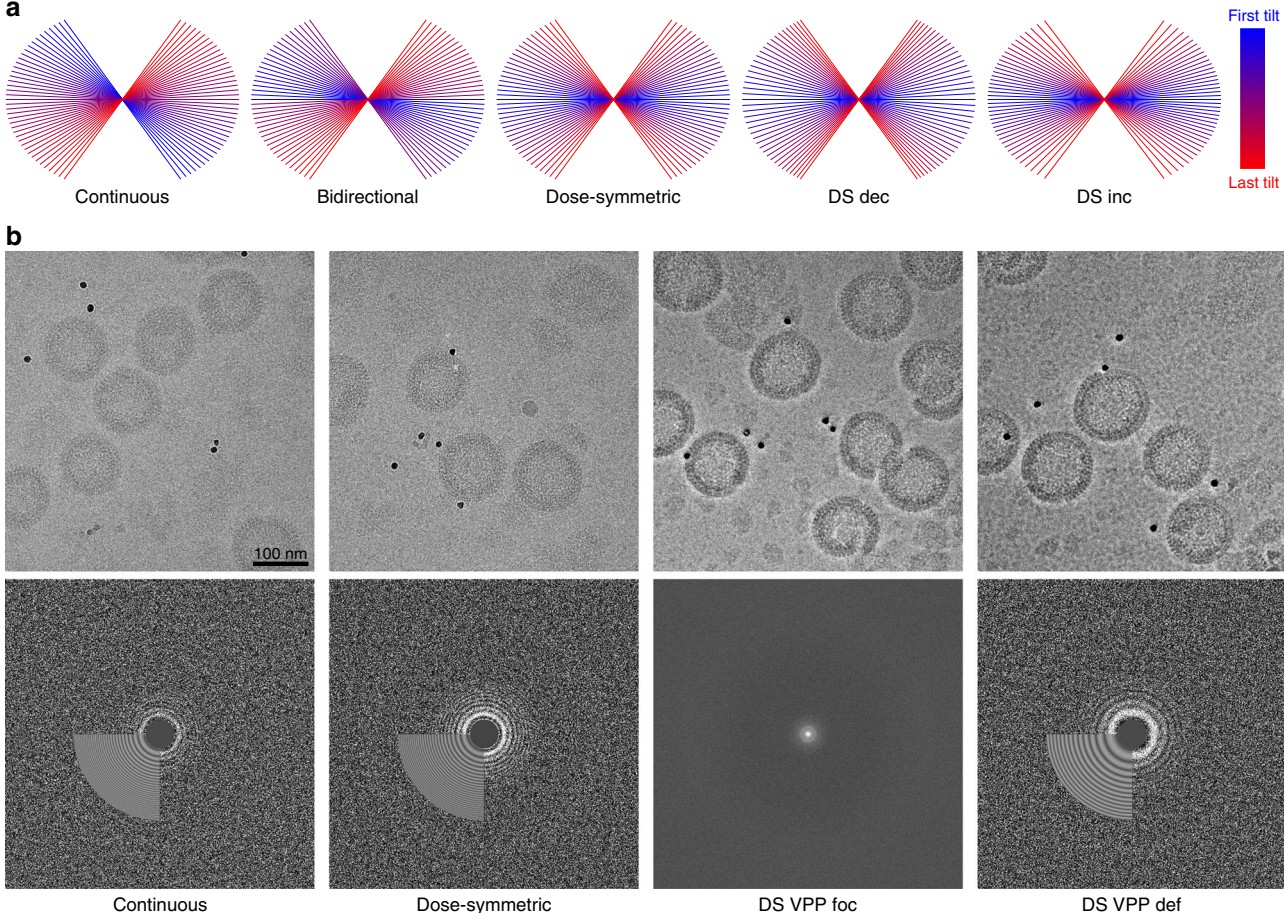

**Fig. 1 Visualization of the acquisition schemes. a** Overview of different angular acquisition schemes used. **b** Zero-degree projections from representative tilt-series and their corresponding periodograms with fitted CTF model estimated by CTFFind4. For DS VPP foc scheme a conventional Fourier power spectrum is shown.

compare the advantage of DS scheme over the other tilt-schemes. Neither have angular increment variations or VPP been systematically tested in combination with DS acquisition. Here, we use the immature HIV-1 lattice as a benchmarking object to systematically study the effect of different acquisition parameters on the resolution attainable by SA. We compare continuous, bidirectional, and DS schemes, each with a constant 3° angular increment; DS schemes with increasing and decreasing angular increment; and DS schemes without and with VPP, both in focus and with underfocus. We find that although each of the schemes is suitable to obtain subnanometer resolution, the DS scheme is indeed the most efficient data collection strategy for obtaining higher resolution that might even be sufficient to build atomic models de novo.

## Results

**Optimal image acquisition comes at the cost of throughput.** We chose the in vitro assembled immature HIV-1 lattice in the absence of BVM as a benchmarking sample, which was originally resolved to 4.5 Å (EMD-4016[14]). We acquired 20–30 tilt-series using seven different acquisition schemes, namely the (i) continuous, (ii) bidirectional, and (iii) DS schemes with even angular increment. To assess the importance of additional acquisition parameters, we further varied the dose-symmetric scheme with (iv) decreasing (DS dec), (v) and increasing (DS inc) angular increment as well as with VPP correction both (vi) in focus (DS VPP foc) and (vii) with underfocus (DS VPP def). The zero-tilt projections together with their periodograms with fitted CTF model from CTFFind4[15] are shown in Fig. 1. The plots indicate

successful CTF fitting and already show the reduced high-resolution information content at 0° in case of the continuous scheme. The VPP projections have high contrast and show the characteristic features in the respective power spectra.

Depending on the specimen, the number of tomograms that can be acquired in a given time frame might be yet another important acquisition parameter because it influences the number of particles in the dataset. The practically achieved, average acquisition time of one tilt-series with 41 projections for each tilt-scheme are shown in Table 1. The continuous scheme is about twice as fast in comparison to the dose-symmetric scheme with VPP in focus. However, the continuous scheme suffers on average from 30% field of view lost; i.e., the position initially selected for acquisition overlaps with the projection acquired at 0° tilt by only 70%. This might be a disadvantage especially for specimen of

**Table 1 Comparison of average times needed for an acquisition of one tilt-series containing 41 images.**

| Tilt-scheme | Average acquisition time per tilt-series |
| --- | --- |
| Continuous | 18 min |
| Bidirectional | 28 min |
| DS | 32 min |
| DS dec | 28 min |
| DS inc | 28 min |
| DS VPP foc | 35 min |
| DS VPP def | 28 min |

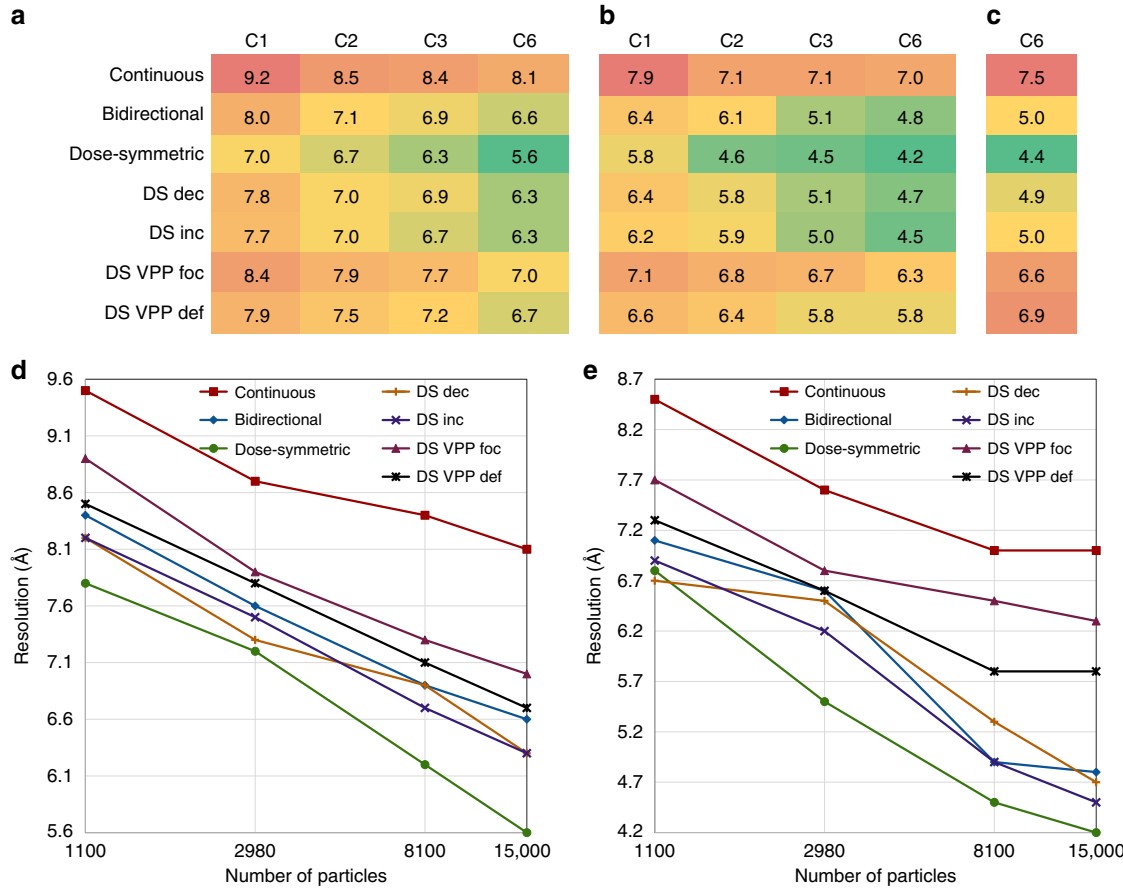

**Fig. 2 Resolution estimates of structures obtained using different acquisitions schemes. a** Resolution between the half maps with different symmetries obtained by FSC at 0.5 criterion. **b** Same as **a**, but the resolution was estimated using the 0.143 criterion. **c** Resolution estimate by FSC (0.5 criterion) between the EMD-3782 map and the respective maps with C6 symmetry applied. The respective FSC curves are shown in Supplementary Fig. 1. **d** B-factor analysis. Plot of resolution of C6-symmetrized structures at 0.5 criterion as a function of number of particles (*x*-axis scaled logarithmically). **e** Same as **d**, but the resolution was estimated at 0.143 criterion.

limited availability, with fewer particles or fiducials. In case of all other acquisition schemes, similar average acquisition times of ~0.5 h per tilt-series were observed. Secondary parameters such as the number of the required focusing or image tracking iterations might have influenced this observation.

**A comparative benchmarking workflow.** All datasets were subjected to a consistent SA workflow including 3D-CTF correction[16], with some deviations that take into account their different nature, i.e., no CTF correction was applied to the VPP dataset acquired in focus (see "Methods" for detail). Since individual tomograms might still differ even in critical properties such as specimen thickness, we implemented a workflow that allows selecting the objectively five best tomograms for each scheme that were then used for benchmarking. Briefly, it uses a multiple sampling approach to find the ideal sub-dataset constellation by optimizing the SA resolution (see "Methods" for detail). To thereby account for variations in VLP content per tomogram, the number of subtomograms contributing to the structural analysis from the five selected tomograms set was set to ~15,000. For detailed overview of parameters and software (SW) used in each step, see Supplementary Table 1.

**Dose-symmetric acquisition is superior.** We aligned each of the structures in multiple iterative rounds of SA (see "Methods"). Since the CA-SP1 is C6 symmetric, we used C1, C2, C3, and

C6 symmetry alignment to systematically assess how dataset size impacts on the attainable resolution. A matrix with the final resolution achieved vs. symmetry is shown in Fig. 2. The overall best resolution of 4.2 Å was obtained with the dose-symmetric scheme with constant angular increment using C6 symmetry (see Fig. 2 and Supplementary Fig. 1). This was measured rather conservatively, with gold standard FSC computed by averaging 5 phase-randomized FSC curves[17]. FSC calculation of our averages against the previously deposited reference structure (EMD-3782) of 3.9 Å resulted in a resolution estimate of 4.4 Å (see Fig. 2c).

Although any of the tested schemes was sufficient to achieve subnanometer resolution, there are considerable differences. While the bidirectional scheme also led to a resolution below 5 Å, the continuous scheme achieved only 7.0 Å—the worst resolution amongst all schemes. Interestingly, the dose-symmetric scheme performs very well already at smaller dataset size. The achieved resolution almost plateaus already at C2 symmetry analysis, while in case of the other schemes it more gradually increases toward C6 symmetry (Fig. 2). In case of the continuous scheme, only a very minor increase in resolution is observed. This is further underscored by B-factor analysis (see "Methods" for details). At FSC 0.5 criterion the resolution increases nearly linearly with the logarithm of the number of particles for all schemes except for the continuous one which starts to flatten already at ~3000 particles (Fig. 2d). This becomes even more apparent at 0.143 criterion (Fig. 2e). While the resolution of the dose-symmetric schemes without VPP still

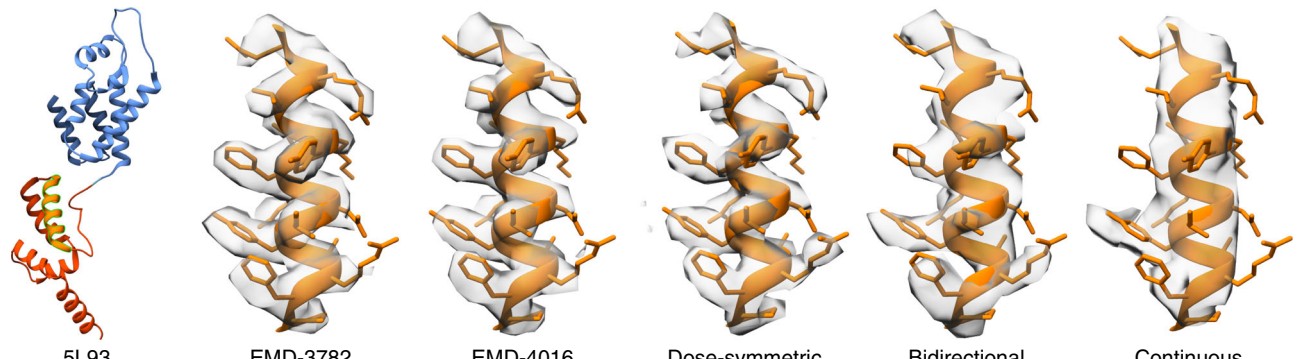

**Fig. 3 Structural details of averages obtained using different acquisition schemes.** An individual helix of HIV-1 CA-SP1 as structurally determined previously is shown in comparison to the dose-symmetric, bidirectional and continuous scheme used in this study. PDB 5L93 indicating the position of the helix is shown left.

increases almost linearly, the other schemes plateau at ~8000 particles.

The bidirectional scheme has also been used with an angular offset[18], meaning that it starts from a given negative tilt angle and increments toward positive angles, while the second branch is shorter and negatively increments toward the minimal angle. The advantage of this scheme is that it continuously acquires projections in a suitable low tilt regime, on the cost of the very first projections that are acquired with a slight tilt. We benchmarked the bidirectional schemes with a starting angle of −20° and 0° against the dose-symmetric scheme (see Supplementary Note 1). We found that both bidirectional schemes perform in a similar way, with only marginal influence of the offset angle (see Supplementary Fig. 3).

In case of the dose-symmetric scheme, we can assess the impact of the angular increment. Although a decreasing angular increment might be beneficial for the resolution of the tomographic datasets[19], these previous considerations were not intended for SA, where the final averages are sampled differently than the initial tomograms. Alternatively, one could argue that an increasing angular increment will distribute less dose toward the high-tilt and large-thickness projections and thus might be superior. At last, uniform angular sampling might be beneficial during the averaging procedure because it simplifies weighting of the angular sampling. We empirically found that indeed the latter is more important. The dose-symmetric tilt-series with varying angular increments resulted in worse resolution than the tilt-series with the constant angular increment. This finding suggests that the increased sampling of high-frequencies on the tomogram level is less important for high-resolution SA than uniform sampling of angles.

Overall, those results are consistent with the observed structural features of respective averages as shown in Fig. 3. The structure obtained from the dose-symmetric tilt-series recovers even more high-resolution features than the equivalent 4.5 Å structure (EMD-4016) from ref. [14], while the 4.8 Å structure corresponding to the bidirectional tilt-series is slightly worse. In all cases, large side chains are very clearly observed. In case of the continuous scheme, even the helical pitch is not discernible clearly.

**Spatial frequency weighting is challenging for VPP datasets.** FSC analysis of the VPP data suggests an overall relatively good performance (Fig. 2). This is particularly remarkable for the dataset acquired in focus, because it had not been CTF corrected. CTF correction is not possible in this case, because the actual function is rather featureless in focus and cannot be reliably fitted. As shown in ref. [16], 3D-CTF correction compensates for defocus variations resulting from the different positions of the individual particles and thus is also relevant for data acquired in focus.

The visual inspection of the respective structures does not credibly support the estimated resolution (Fig. 4). The typical high-resolution features are not observed, suggesting inaccurate spatial frequency weighting. A variation of averaging parameters such as high-pass filtering or sharpening with different arbitrarily chosen B-factors did not recover the respective structural features. Amplitude matching using the 4.2 Å structure from the dose-symmetric scheme as a reference only partially resolved these issues (see Supplementary Fig. 2). We conclude that although the respective high-resolution information might be contained in the average, it is nontrivial to recover de novo. The underfocused VPP dataset, although 3D-CTF corrected, suffers from the same problem. One might thus speculate that high-pass filtering at the SA level is insufficient and different filters might be rather used already during the tomogram reconstruction in order to suppress the very pronounced low frequencies.

**Tomogram alignment accuracy impacts on the resolution.** The importance of the accuracy of the alignment of the projections for SA has often been argued but to the best of our knowledge, not yet been systematically quantified. To test the influence of the tilt-series alignment precision on the attainable resolution for the given benchmarking dataset, we introduced errors to the fiducial-based alignment models by artificially adding shifts into a random direction in sillico (only for the dose-symmetric scheme). We reconstructed the respective tomograms and proceeded with SA workflow starting with 4× binned tomograms (assuming that the errors would not have a significant impact on 8× binned data) with C1 and C6 symmetry. The results are summarized in Table 2. While error of 0.5 pixels is negligible for both C1 and C6 symmetry the impact of displacement by 2.0 pixels seems to be more significant for structures with high-resolution worsening the resolution by 0.9 Å and 1.4 Å, respectively. This is in line with the residual error and its standard deviation reported by eTomo[20] during the alignment routine—a shift by 0.5 pixels does not significantly increase the residual error nor the standard deviation while a shift by 2.0 pixels increases the residual error by factor of 2.

Recently, a method has been put forward that corrects for local distortions that might arise during the exposure (due to beam-induced specimen movement)[21]. Thereby the subtomogram positions that were refined by averaging are treated as fiducials to locally improve the tomogram reconstruction. We tested this workflow on the data presented here and found only very minor improvements (see Supplementary Note 2). This might be either due to the preselection of most suitable tomograms from the overall data or due to the dynamic dose filtering, which effectively

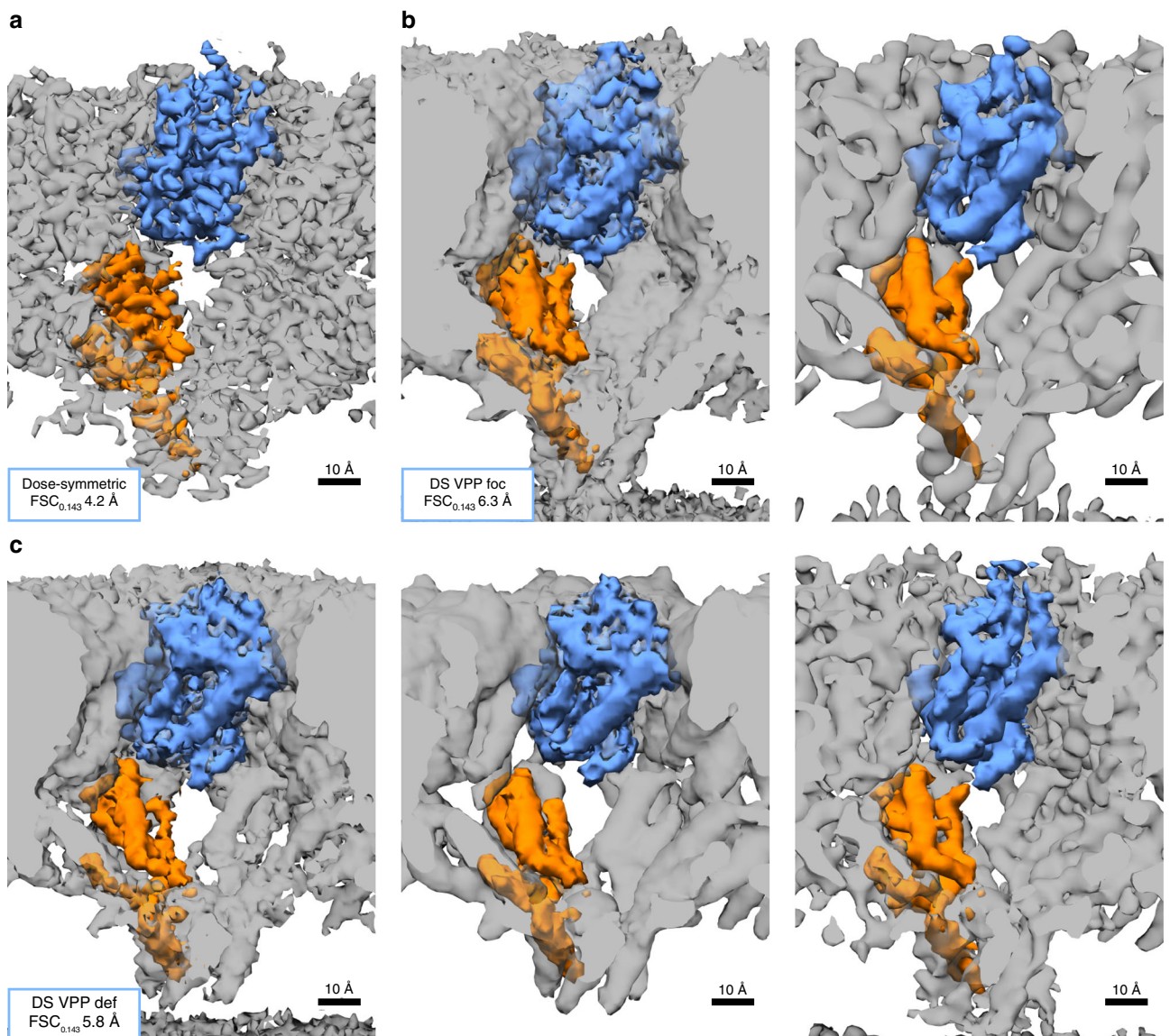

**Fig. 4 Cryo-EM maps of HIV-1 CA-SP1. a** Structure obtained by the dose-symmetric scheme (after CTF-reweighting and sharpening). **b** A raw structure obtained from DS VPP foc scheme and its sharpened version (right). **c** Raw structure obtained from DS VPP def scheme, corresponding CTF-reweighted structure (middle) and final structure after CTF-reweighting and sharpening (right). All images are color coded according to a single chain from PDB 5L93 of HIV-1 CA-SP1 (see Fig. 3 for comparison).

**Table 2 Influence of the tilt-series alignment precision on final resolution of the structure solved from the dataset obtained from DS scheme.**

|  | Average residuals | Average STDs | Resolution at 0.5 for C1 | Resolution at 0.5 for C6 | Resolution at 0.143 for C1 | Resolution at 0.143 for C6 |
|---|---|---|---|---|---|---|
| Original | 0.59 | 0.35 | 7.0 | 5.6 | 5.8 | 4.2 |
| 0.5 pixels error | 0.65 | 0.36 | 7.2 | 5.7 | 5.6 | 4.2 |
| 1.0 pixels error | 0.81 | 0.44 | 7.4 | 6.1 | 6.0 | 4.6 |
| 1.5 pixels error | 1.01 | 0.48 | 7.7 | 6.3 | 6.3 | 4.7 |
| 2.0 pixels error | 1.23 | 0.57 | 8.3 | 6.7 | 6.7 | 5.6 |

removes high-resolution information from projections that have been exposed to a higher dose.

## Discussion

During the last decade, CryoET has gained enormous momentum and has become an important method to structurally analyze macromolecules in their native context. However, the aspect of how to optimally acquire the data has remained somewhat unorganized. Here, we have systematically compared different tomographic tilt-schemes in order to lay down a path toward high resolution to SA. Under the experimental conditions we chose for our benchmarking study, the dose-symmetric scheme with the constant angular increment outperformed all other tested

**Table 3 Image acquisition parameters that differ for each of the benchmarked schemes. The initial tilt-step for DS dec and DS inc scheme was determined in a way that the whole tilt-series contained 41 images.**

| Tilt-scheme | Tilt-step | Acquisition order | Defocus range | Defocus step |
|---|---|---|---|---|
| Continuous | 3° | −60, −57, ..., 0, ..., 57, 60 | −1.5 −4.0 μm | 0.25 μm |
| Bidirectional | 3° | 0, −3, −6, ..., −60, 3, 6, ..., 60 | −1.5 −4.0 μm | 0.25 μm |
| Dose-symmetric | 3° | 0, −3, 3, 6, −6, ..., 60, −60 | −1.5 −4.0 μm | 0.25 μm |
| DS dec | initialStep × cos(currentStep); initialStep = 3.7° | 0, −3.7, 3.7, ..., 57.9, 59.9 | −1.5 −4.0 μm | 0.25 μm |
| DS inc | initialStep/cos(currentStep); initialStep = 2.5° | 0, −2.5, 2.5, ..., 54.8, 59.1 | −1.5 −4.0 μm | 0.25 μm |
| DS VPP foc | 3° | 0, −3, 3, 6, −6, ..., 60, −60 | – | – |
| DS VPP def | 3° | 0, −3, 3, 6, −6, ..., 60, −60 | −1.0 −3.0 μm | 0.25 μm |

schemes in terms of ultimately obtained resolution. While the bidirectional scheme, with or without offset, provides a reasonable alternative in terms of acquisition time to resolution ratio, the continuous scheme has clear limitations. Despite superior acquisition speed, our results clearly suggest that even with twice the number of particles the resolution does not further improve beyond the 7 Å regime. Also variations of the angular increment were not beneficial. However, the question of the optimal angular increment (together with nonconstant dose distribution within the tilt-series) was not addressed in this study and most likely will be sample dependent. Although the differences in the final resolution attained might not seem tremendous, they can be of critical importance if a structure is determined de novo.

The acquisition and analysis of VPP datasets comes with additional challenges, such as VPP conditioning, stability, increased acquisition time as well as phase-shift and defocus determination, heavily oversampled low frequencies and others. As far as we can see, there is no clearly defined way to recover the high-resolution features for a given structure de novo, and even if so, the resolution was comparably lower. We thus conclude that although VPP data are sufficient to obtain subnanometer resolution, they are not beneficial for maximizing resolution in SA. The better contrast however might be beneficial for the identification of particles in cases where high resolution is not required. The high contrast of VPP imaging is highly beneficial for cellular, biological, and ultrastructural investigations, however, further work is required to unlock its full potential for SA analysis.

The 3D maps of final structures with C6 symmetry as well as their corresponding half maps (both raw and CTF-reweighted) are publicly available at EMDB (EMD-10207) and the raw tilt-series together with all files relevant for SA are available at EMPIAR (EMPIAR-10277) and can be used to further develop and/or benchmark processing routines for SA. In addition to the presented data the EMPIAR deposition also contains 8 tilt-series acquired at regions without any gold fiducials. We hope these tilt-series will be used to test and improve current fiducial-less alignment techniques.

We believe that our conclusions are generic for projects where particle number, specimen thickness, angular coverage, and available fiducials are not limiting. To which extent they are applicable to thicker or fiducial-less specimen, such as e.g., obtained during FIB-SEM projects, remains to be tested in the future.

## Methods

**Sample preparation**. The sample of HIV-1 ΔMACANCSP2 VLPs was prepared as described in[14]. Degassed (i.e., stored in a vacuum desiccator) 2/1–3C C-flat grids were glow discharged for 45 s at 20 mA (using Ted Pella Pelco EasyGlow discharger). VLP solution was diluted with 10 nm colloid gold (obtained from Utrecht university—http://www.cellbiology-utrecht.nl/products.html) in VLP sample buffer and 2.5 μl of the solution was applied to the grids and plunge frozen in liquid ethane using FEI Vitrobot Mark III at the temperature of 15 °C and relative humidity of ~90% (blotting time 1.0 s). The blotting paper used was Whatman 597.

**Image acquisition**. To minimize biological variations of the sample, all datasets were collected on the same grid. All datasets were collected on FEI Titan Krios TEM at 300 keV, with dual-axis holder and Gatan K2xp direct electron detector using a Quantum LS 967 energy filter with slit width of 20 eV. Projections were acquired using SerialEM SW[22] as 4 K × 4 K movies of 10–20 frames in the counting mode at the magnification of 105,000× which corresponds to the pixel size of 1.33 Å. The frames were aligned using MotionCorr[23]. For all datasets the tilt range was ±60° with 41 projections per tilt-series, with constant exposure time and target total dose of ~140 e per Å$^2$ (corresponds to an incident dose of ~3.5 e per Å$^2$ per projection). The overview of parameters that differ among the schemes is shown in Table 3. The continuous scheme was collected with the SerialEM tilt controller using parameters shown in Supplementary Table 2. All other tested schemes were collected using drift measurements and stage tilt backlash as described in ref. [12]. We collected 20–30 tilt-series for each scheme.

**Image processing**. Step 1. Initial pre-processing: for all tilt-series, we performed CTF estimation using CTFFind4 and corrected for dose-exposure as described in ref. [24] using Matlab implementation that was adapted for the tomographic tilt-series[25]. Tilt-series that contained one or more inadequate tilt-images (i.e., not properly tracked or failed CTF estimation) were discarded. For the following steps eTomo[20] was used. The pixels with outlier intensities were removed and preliminary alignment was computed based on cross-correlation. The automatic seeding procedure was used to find the gold fiducials for alignment and the seeding model was manually corrected such that it contains only fiducials that are present in the field of view in all projections (on average four to five fiducials per tilt-series fulfilled this constraint). The tilt-series with less than three fiducials were eliminated from further processing. The fiducials were automatically tracked and in cases where tracking failed the model was corrected manually. The fiducial centers were manually refined prior the final alignment. Tomograms were reconstructed 8× binned and using SIRT-like filter (except for DS VPP foc and DS VPP def datasets, as their contrast was sufficient using radial filtering). The tomograms were used to position the center of mass into the center of tomogram along z axis as well as to assess tomograms thickness and the quality of the alignment—all tilt-series where the fiducials showed strong movement in tomograms were removed from further processing. From the remaining tilt-series, the most suited 8–10 tilt-series per dataset were chosen for further processing based on the alignment residuals, defocus range, and specimen thickness. The strict selection criteria were used to eliminate potential quality differences as much as possible.

Step 2. Tomogram reconstruction: tomograms were reconstructed with 3D-CTF correction using novaCTF[16]. Multiplication was used as the correction method, with 15 nm slab size and astigmatism correction. The DS VPP foc dataset was also reconstructed using novaCTF with the CTF correction turned off. To ensure accurate phase-shift estimation, the DS VPP def tomograms were reconstructed both with and without 3D-CTF correction. The uncorrected tomograms were used until step 5. Tomograms were subsequently binned 2×, 4×, and 8× using Fourier cropping.

Step 3. Particle picking: similar to ref. [14], the centers of the VLPs were picked manually and their spherical shape was used to generate initial positions and orientations on the lattice[26]. The lattice was oversampled, i.e., on average 10× more positions were created than assumed number of subunits. The center picking was done in IMOD on the 8× binned tomograms from step 1, i.e., reconstructed using SIRT-like filter. These tomograms were used only to generate list of positions, for SA itself the tomograms reconstructed using novaCTF, as described in step 2, were used. The particles were picked not only from perfectly preserved VLPs (or VLPs that were fully in the field of view), but also from the incomplete VLPs. The precision of the center picking is not crucial for the quality of the final structure—already in the first two iterations of alignment, the initial positions shift to the lattice.

Step 4. Reference creation: for each dataset one tomogram was chosen (typically the one most underfocused and thus with strong low-frequency information) that was used to create the initial reference. For DS VPP foc dataset a reference was created from each tomogram and the one visually closest to the references from other datasets was chosen. Twenty iterations of alignment were run to obtain a reference for each dataset. All starting references were shifted and rotated to have

the same position and orientation within the box to facilitate further processing (e.g., same masks could be used for all the datasets) as well as structural analysis.

Step 5. SA: two iterations of alignment were run on particles from 8× binned tomograms using the references obtained in the previous step. At this stage misaligned particles were discarded. This was done fully automatically, using ellipsoid fitting and removing particles that deviated above the standard deviation either in angle or in radius (see Supplementary Note 3 for more details). So-called distance cleaning was performed—particles that shifted to the same position were also discarded (the criterion for choosing the better particle was angular distance based on the ellipsoid fitting). Approximately 8% of particles were left for each dataset which given the oversampling of initial positions corresponds to roughly 80% of actual subunits. The subsequent SA workflow exactly followed the protocol from ref. [14].

For DS VPP def dataset this step was still done using particles from the tomograms without 3D-CTF correction and the final positions and orientations were subsequently used to generate an average using particles from the 3D-CTF corrected tomograms. The improvement in resolution w.r.t. the uncorrected structure confirmed an accurate phase-shift estimation and the corrected tomograms were thus used for all subsequent processing steps.

Step 6. Selection of five best tomograms from each dataset: for each dataset, all possible combinations of five tomograms were generated and an average structure for each of the combinations was computed using the orientations and positions from the final alignment of unbinned particles. Each tomogram within the dataset contributed with the same amount of particles (particles were randomly removed from each tomogram to match the tomogram with the least number of particles). Resolution at 0.143 was computed and the subset with the best resolution for each dataset was chosen for further processing.

Step 7. Reconstruction of five tomograms subsets: the final positions from the SA alignment (step 5) were used to compute the center of mass for each tomogram and all tomograms from the chosen subsets were reconstructed using novaCTF with the refined underfocus shift. For DS VPP foc dataset this step was omitted.

Step 8. SA workflow of five tomograms subsets: for each dataset the step 4 was repeated, creating a reference using one of the tomograms from the subset. Two iterations of alignment were run on particles from 8× binned tomograms followed by ellipsoid-based removal of misaligned particles and distance cleaning (see Supplementary Table 3). All VLPs with more than 50% of particles removed during the ellipsoid-based cleaning were discarded from further processing. From the remaining particles a random selection was removed and the alignment continued with ~15,000 particles. The subsequent SA workflow at lower binning exactly followed the protocol from[14]. For unbinned particles four iterations of alignment were run (see Supplementary Table 1).

Step 9. Testing the influence of the number of particles: two approaches were used to assess the influence of the number of particles on the final structure and attainable resolution. First, we exploited the symmetrical property of the structure. Step 8 was repeated for each dataset using C1, C2, and C3 symmetry, effectively reducing the number of particles 6×, 3×, and 2×, respectively. Second, we used B-factor analysis as proposed in ref. [27]. For each dataset, three logarithmically smaller subsets of particles were randomly selected from the final set of particles (i.e., 1100, 2980, and 8100). For each of the subset, three iterations of alignment were run on unbinned data using the positions and orientations obtained in step 8 as a starting point. This analysis was done using C6 symmetry.

**Reporting summary**. Further information on research design is available in the Nature Research Reporting Summary linked to this article.

## Data availability
The 3D maps are available at EMDB database (accession code EMD-10207) and the raw tilt-series at EMPIAR (accession code EMPIAR-10277).

## Code availability
All unpublished code used in this study is available upon request.

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

## Acknowledgements
We thank Drs Matteo Allegretti, Julia Mahamid, Jürgen Plitzko, Florian Schur, and William Wan for helpful discussions. H.G.K. and M.B. acknowledge funding by the German Research Association (DFG, project number 240245660; project 5). M.B. acknowledges funding by EMBL, the Max Planck Society and the European Research Council (#724349 ComplexAssembly). Christian E. Zimmerli is a candidate for joint PhD degree from EMBL and Faculty of Biosciences, Heidelberg University.

## Author contributions
B.T., W.J.H.H. and M.B. conceived the project. M.B. and H.G.K. supervised the project. M.O. performed the sample preparation and prepared cryo-EM grids. W.J.H.H. and B.T. collected the data. B.T. developed and performed the pipeline for the data analysis. B.T. and S.M. evaluated the results. J.W.B. implemented under supervision of C.E.Z. the script for correction of local distortions. B.T. and M.B. wrote the manuscript with input from W.J.H.H., M.O., S.M., and H.G.K.

## Competing interests
The authors declare no competing interests.
