## [Peer Review File · Nature Communications]

Reviewers' Comments:

Reviewer #1:

Remarks to the Author:

The authors present an analysis of tilt-series collection strategies for cryoET. The analysis claims that the dose symmetric scheme produces higher-resolution sub-volume EM maps than any other scheme tested. This appears to be the first systematic analysis of collection schemes at sub-nanometer resolution and so would have a moderately-high impact on the field of cryoET. However, the analysis may be significantly or critically incomplete pending further investigation by the authors.

There is one major concern I have that the authors have not dealt with nor mentioned and which may significantly change the analysis and so must be addressed:

-At least one more important technical reason for resolution limits in sub-tomogram averaging exists: specimen and ice movement in 3D during exposure. Ice movement includes doming of the ice in the direction perpendicular to the grid. Beam-induced specimen movement in 3D results in projections of effectively different sets of objects due to translational and rotational specimen movements. Different specimen, different grid types, different collection properties (e.g. dose rate), and different collection schemes directly affect how much 3D specimen and ice movement exists in each tomogram. These 3D movements may be alleviated by constrained per-particle sub-tilt-series refinement, such as with emClarity or EMAN2, as long as enough combined SNR in the sub-tilt-series exists. The study presented by the authors may be affected by 3D specimen and ice movement, however this was not even considered in the analysis. For example, if this is an issue then bi-directional tilt-series collection is likely to result in poorly-aligned tilt images between the neighboring tilt images from the first and second half of the tilt-series due to disjointed object locations in 3D. This may be a primary reason why bi-directional collection is significantly worse than dose-symmetric; dose-symmetric tilt-series do not have neighboring tilt images from disjointed object locations in 3D. Since only global tilt-series is performed in this study, then this affect has not been decoupled from the analysis and so needs to be investigated directly because this is a central claim of the study. The authors acknowledge a sharp decline in information between the neighboring tilt images of the two bi-directional halves (page 3, line 12), but not 3D movement. To address this point directly, the authors could perform sub-tilt-series refinement. However, this might not currently be compatible with NovaCTF correction. To partially address this point in silico, the authors could add increasing systematic errors (ie. non-random) to fiducials to attempt to mimic bi-directional 3D specimen and ice movement in dose-symmetric tilt-series. However, this is unlikely to be conclusive. Nevertheless, this problem needs to be addressed in the manuscript.

I have two moderate concerns that is connected to the above major concern:

-One collection method that is commonly used in many labs is an offset bi-directional scheme where most low-tilts are collected first. For example, [-20:60] then [-20:-60]. This scheme may also ameliorate the major concern above, and so should at least be mentioned if not tested.

-In the first paragraph of the introduction, at least one more prerequisite for obtaining high resolution exists: orientational distribution of the biological objects should cover all of Fourier space. Many specimen have limited orientational distributions. The specimen used in the manuscript does not have this issue, which should be emphasized. This additional prerequisite limits the applicability of the DS dec and DS inc studies presented, as they may be more useful for objects with strong preferred orientation.

I also have several minor concerns and comments/edits:

-It is unclear how the sentence beginning on page 7, line 5 relates to in focus VPP collection, which is the topic of the paragraph; ie. Since there is no defocus estimation, how are the defocus offsets of particles taken into account? Is the defocus estimated to be zero or slightly above zero - this is not clear.

-Add a scale bar to Figure 4 and Figure S2.

-How were the super-resolution movies binned? In Fourier space?

-Page 9, line 28: 'constrain' > 'constraint'

-Why were incomplete VLPs also picked? This seems like an additional unnecessary variable that may affect resolution due to structural variation.

-If the authors haven't already, consider also depositing processed data and processing folders to EMPIAR. You can simply ask EMPIAR staff to let you upload more data to the already-existing entries.

Reviewer #2:

Remarks to the Author:

This manuscript addresses an important question in cryo-ET data collection - what's the best data collection scheme for subtomogram averaging. The insight provided by the systematic analysis will be of great interest to the growing cryo-ET user community. Considering its potential high impact, the current version needs clarification and updates.

(1) Methods-Image Acquisition says only continuous scheme was collected using the built-in function in SerialEM. SerialEM has built-in function for bidirectional too (TS setup-Run series in two directions). Why the authors chose not to use SerialEM's built-in function? Please compare the throughput of the built-in function with the customized script.

(2) Besides the schemes tested in this manuscript, another modified bidirectional scheme is worth investigating. In this scheme, the first tilt starts at intermediate tilt angles, say 30 deg. From +30 deg, it tilts toward 0, passing 0 and continuing toward negative angles until reaching -60deg. From +30 to -60, an image is recorded every 3deg (31 tilts in this first part). After that, it goes back to 30 deg, continuing toward +60deg and taking one image every 3deg (10 tilts in this second part). By doing this, the relatively less damaged data were collected at relatively low tilt angles. It is a compromise between throughput and retaining the high resolution information at low angles. I suggest the authors add this to their comparison. One complication is that all the existing datasets are collected on one single grid, and it may not be possible to use the same one for this additional test. Nevertheless, the results can enhance the completeness of the manuscript.

(3) Recently, SerialEM's development version incorporates built-in functions for dose-symmetric as well. More users probably would prefer to use the built-in function. The authors should benchmark the built-in DS scheme using parameters as close to their native Hagen scheme as possible and report if the built-in DS scheme gives a higher throughput or other advantages. Update Table 2 accordingly. Ref: <http://bio3d.colorado.edu/ftp/SerialEM/History-beta.txt>

(4) Methods say 20~30 tilt series per scheme were collected, but only 8~10 tilt series were used for further processing. Is it common practice in the authors' lab to throw away >50% of raw data at this stage? Please comment on this practice in the manuscript, to give the audience an idea about the importance/necessity of getting rid of poor data early on.

(5) Related to (4), page 11 says 8% of picked particles were left. Please provide more details on the

criteria used for the clean up (quantitatively if possible). How sensitive is the final result to this step? Will slightly more/less strict criteria make a big difference?

(6) Among the software used (Table S1), some are common, such as eTomo and CTFFind, while others are not, such as Dose Exposure correction (matlab scripts), and novaCTF. Please comment in the manuscript on the importance of using these less common tools and the computational cost, whether they should be used routinely for all subtomogram projects, or used selectively for specific projects. I understand each of them has its own publication, but it would be helpful to have a few comments for the audience to grasp the (critical) roles they play in the process and whether they should be widely adopted in the cryo-ET community.

(7) Table S2 says a 10s delay is used. Is it really necessary? For 41 tilts, that's almost 7min. Is 10s an empirical number or the authors have done some tests and found 10s gives the optimal results?

(8) Table S2 says Beam intensity is kept constant, is exposure also kept constant or varied. I suggest the authors replace Table S2 with a screenshot from SerialEM Tilt series Setup, so that it's easy to follow which parameters are used.

(9) Table 1 shows a defocus gradient is used. How is this achieved? SerialEM's TS setup only allows one defocus value to be specified. When SerialEM's built-in TS function is used, did the author use a script or just manually change the TS setup every few series? Please comment on this in the manuscript. My personal experience is that even when a script is used in the initial-task to vary the target defocus, SerialEM only takes the value from the TS setup. This seems to be a hurdle for automated data collection.

(10) Methods-Acquisition. Please specify the type of holder, the dual axis holder typically on the Krios G2, or the single axis holder on G3? I'm wondering if the (presumably more stable) single axis holder alleviates the problem of losing field of view.

(11) Methods-Acquisition. Please specify the version of SerialEM, as specific as possible. SerialEM undergoes constant changes. Version number is necessary for reproducibility purposes. Also add the version of GMS (DM). These information may be incorporated into Table S1.

(12) Have the authors attempted to increase the throughput utilizing the FrameSeriesFromVar function in SerialEM? This function allows the user to potentially skip focusing and/or tracking steps by feeding a list of calibrated focus/image shift values to the program. FEI tomography offers this option for years. Ref: http://bio3d.colorado.edu/SerialEM/betaHlp/html/script_commands.htm

(13) For in-focus VPP dataset, did the authors go the extra mile to improve the accuracy of focusing, such as iterating the defocus measurement at opposite sides? Table 2 shows 'DS VPP foc' takes the longest time.

(14) Page7, L7: "The visual inspection of the respective structures does not credibly support the estimated resolution (Fig 4)". Please add the nominal resolution as insets to each panel in Fig. 4. Add a panel of the density map from bi-directional scheme to Fig4. Line2 says the FSC shows VPP result is similar to bi-directional. It is helpful to see a comparison of the density maps of the two schemes.

(15) Methods-Sample preparation. What does 'degassed' mean? How is it performed? Add the type of instrument used for glow discharge. Add the supplier of the colloid gold. Add the type of blotting paper.

(16) Page 10, particle picking. Add the initial number of particles.

(17) Page 8, second paragraph essentially says VPP is not a good idea for subtomogram averaging aiming for high resolution, given the tools available now. Other cryo-ET scientists have given similar advice for years. Now we finally have published results to support it. I feel the authors can strengthen the conclusion, giving the audience a clearer message.

Terminology issues:

1) Minimum tilt angle (e.g. page 3, L6). I believe it refers to the most negative tilt angle, which is actually a high tilt angle. Calling it minimum is ambiguous. Change it to the most negative tilt angle or something less ambiguous.

2) The lowest defocus (page 10, L7). Based on the context, I believe it refers to the most underfocus value. Defocus means out of focus, which can be over- or under-focus. When I hear low defocus, I relate it to close to focus. Change it to the highest underfocus or something less ambiguous.

Minor correction:

Page2, line 8: The most important properties are ...

Page 12, line 5: For each dataset, 3 logarithmically smaller ...

We want to thank the reviewers for the very constructive criticism that was very helpful to improve our manuscript. On a general note, we want to point out the major objective of our manuscript is to investigate how the distribution of dose across the projections of a tomogram affects the attainable resolution in subtomogram averaging. We believe we have thoroughly addressed this point under well controlled conditions. Additional parameters such as dynamic exposure times, alternative tilt schemes, the impact of preferred orientation, alternative software solutions for data acquisition or processing are challenging to optimize systematically, given computational and beam time resources available to us. We have nevertheless followed several of the reviewer's suggestions to include additional data and analysis such as i.e. an assessment of the bidirectional tilt scheme with offset. Our detailed point by point response follows below.

Reviewers' comments:

Reviewer #1 (Remarks to the Author):

The authors present an analysis of tilt-series collection strategies for cryoET. The analysis claims that the dose symmetric scheme produces higher-resolution sub-volume EM maps than any other scheme tested. This appears to be the first systematic analysis of collection schemes at sub-nanometer resolution and so would have a moderately-high impact on the field of cryoET. However, the analysis may be significantly or critically incomplete pending further investigation by the authors.

There is one major concern I have that the authors have not dealt with nor mentioned and which may significantly change the analysis and so must be addressed:

-At least one more important technical reason for resolution limits in sub-tomogram averaging exists: specimen and ice movement in 3D during exposure. Ice movement includes doming of the ice in the direction perpendicular to the grid. Beam-induced specimen movement in 3D results in projections of effectively different sets of objects due to translational and rotational specimen movements. Different specimen, different grid types, different collection properties (e.g. dose rate), and different collection schemes directly affect how much 3D specimen and ice movement exists in each tomogram. These 3D movements may be alleviated by constrained per-particle sub-tilt-series refinement, such as with emClarity or EMAN2, as long as enough combined SNR in the sub-tilt-series exists. The study presented by the authors may be affected by 3D specimen and ice movement, however this was not even considered in the analysis. For example, if this is an issue then bi-directional tilt-series collection is likely to result in poorly-aligned tilt images between the neighboring tilt images from the first and second half of the tilt-series due to disjointed object locations in 3D. This may be a primary reason why bi-directional collection is significantly worse than dose-symmetric; dose-symmetric tilt-series do not have neighboring tilt images from disjointed object locations in 3D. Since only global tilt-series is performed in this study, then this affect has not been decoupled from the analysis and so needs to be investigated directly because this is a central claim of the study. The authors acknowledge a sharp decline in information between the neighboring tilt images of the two bi-directional halves (page 3, line 12), but not 3D movement. To address this point directly, the authors could perform sub-tilt-series refinement. However, this might not currently be compatible with NovaCTF correction. To

partially address this point in silico, the authors could add increasing systematic errors (ie. non-random) to fiducials to attempt to mimic bi-directional 3D specimen and ice movement in dose-symmetric tilt-series. However, this is unlikely to be conclusive. Nevertheless, this problem needs to be addressed in the manuscript.

We agree that distortions cause by beam induced movement might influence the attainable resolution. We further agree that it might seem intuitive that local distortions would impact more onto the bidirectional scheme as compared to others. In practice however, this is not the case, as we demonstrate below.

The tilt-series alignment procedure in eTomo is not affected by the order in which the projections are acquired. It can be split in two parts, fiducial tracking and alignment computation. The fiducial tracking aims to find fiducial projections through the whole tilt-series to create its trajectory. The search area is based on a position in the previous tilt and a tilt angle. Thus, if there is a considerable image shift between neighboring tilts the tracking often fails. This indeed happened more often for the bidirectional scheme. It however did not have any influence on the final outcome as we corrected for it manually. The final alignment computes transformation for each projection by minimizing distances of all fiducials to their respective trajectories. Thereby, the contribution of fiducials is “blind” to both, the order of projections and their tilt-angle. Thus, the beam induced movement will influence the global alignment of all schemes the same way.

The fact that local distortions accumulate together with the dose spent, is counteracted by dose-filtering. Thereby, dynamic low pass filtering reduces the contribution of (higher dose) projections that might have larger distortions to the final averages.

We have meanwhile implemented local corrections in our lab (see next paragraph below) and obtained encouraging results with ribosomes that comprise very contrast-rich objects. The asymmetric unit of HIV capsid is way smaller though. We are very aware that Himes et al successfully improved the resolution outcome as compared to Schur et al using the very same specimen. We however want to remind the reviewer that Himes et al at the same time made various other improvements, one of which being to consider a much larger set of asymmetric units. The contribution of these individual improvements to the overall increased resolution are thus difficult to disentangle.

As suggested, we performed the sub-tilt-series alignment on the dose-symmetric and the bidirectional dataset using the method described by Himes et al. The final averages together with their positions and orientations were used to create new fiducials. A new alignment was computed in eTomo using local alignments. Since novaCTF does not currently support the local alignments, this analysis was done using tomograms corrected with 2D-CTF strip-based approach from IMOD (ctfphaseflip). The positions and orientations from the final alignments were used to directly compute the final average from 2D-CTF corrected tomograms, and lead to worse resolution as compared to 3D-CTF correction, as expected. A new set of 2D-CTF corrected tomograms was reconstructed using the local refinements, followed by SA. The obtained nominal resolution was the same as without the local alignments for the dose-symmetric scheme and improved only marginally for the bidirectional scheme (see FSC curve below).

We thus conclude that correcting for local distortions does not have a significantly different impact on the bidirectional scheme than on the dose-symmetric. We discuss this issue in the manuscript on page 8, line 10.

I have two moderate concerns that is connected to the above major concern:

-One collection method that is commonly used in many labs is an offset bi-directional scheme where most low-tilts are collected first. For example, [-20:60] then [-20:-60]. This scheme may also ameliorate the major concern above, and so should at least be mentioned if not tested.

This is a very valid point, which has been raised by both reviewers. As suggested, we acquired additional bidirectional dataset starting with starting tilt-angle -20° together with a control dataset using the dose-symmetric scheme (both on the same grid, but a new grid). The results are shown in the supplementary, section 1. The resolution obtained with the control dataset (dose-symmetric) was the same as for the original dose-symmetric dataset (i.e. 4.2Å), while the bidirectional scheme with starting tilt-angle offset resulted in a structure with resolution of 4.7Å – thus the improvement over the bidirectional scheme with 0 starting tilt-angle is marginal (~0.1 Å). The B-factor analysis further underlines that the differences between both bidirectional schemes relatively minor. We discuss this in the main text of the revised manuscript (page 6, line 18).

-In the first paragraph of the introduction, at least one more prerequisite for obtaining high resolution exists: orientational distribution of the biological objects should cover all of Fourier space. Many specimen have limited orientational distributions. The specimen used in the manuscript does not have this issue, which should be emphasized. This additional prerequisite limits the applicability of the DS

dec and DS inc studies presented, as they may be more useful for objects with strong preferred orientation.

We agree that the angular coverage or preferred orientation might limit the resolution and have included a respective remark into the introduction (page 2, line 12 and 28). Sampling more densely in a specific region would not help with preferred-orientation – the problem is the missing wedge which remains the same (or similar) for the subtomograms with the same orientation.

I also have several minor concerns and comments/edits:

-It is unclear how the sentence beginning on page 7, line 5 relates to in focus VPP collection, which is the topic of the paragraph; ie. Since there is no defocus estimation, how are the defocus offsets of particles taken into account? Is the defocus estimated to be zero or slightly above zero - this is not clear.

We wanted to emphasize that even for datasets collected in focus one should still correct for CTF modulations arising from shifts in Z positions due to specimen thickness and tilting. The benefit of this has been proven for this particular sample (Turanova et al. 2017). This is however not possible for the VPP dataset acquired in focus, because the actual function is rather featureless in focus and cannot be reliably fitted. We rephrased the paragraph to make it more clear (page 7, line 11).

-Add a scale bar to Figure 4 and Figure S2.

Thank you for pointing that out – we added the scale bars to both figures.

-How were the super-resolution movies binned? In Fourier space?

We did not collect 8K x 8K super-resolution movies as previously stated in the manuscript but 4K x 4K movies in the counting mode (and thus no binning was necessary). We corrected this in the manuscript. (page 10, line 5).

-Page 9, line 28: 'constrain' > 'constraint'

Thank you for noticing – we corrected it.

-Why were incomplete VLPs also picked? This seems like an additional unnecessary variable that may affect resolution due to structural variation.

The virus like particles (VLPs) are not real viruses but in vitro assembled from recombinant protein and membranes. We used the term 'incomplete' because the respective lattices are almost never fully closed, which is simply a consequence of the preparation protocol. This should however not influence the final structure as the geometric-based cleaning reliably removes badly aligned particles even for both complete and incomplete VLPs. We added a section to the Supplementary that explains the cleaning procedures in more detail and includes figures showing the automatic cleaning results.

-If the authors haven't already, consider also depositing processed data and processing folders to EMPIAR. You can simply ask EMPIAR staff to let you upload more data to the already-existing entries.

In addition to all tilt-series the EMPIAR deposition also includes all files necessary to reproduce our results, such as alignment transformation from eTomo, tomograms' dimensions and shifts, positions and orientations of final subtomograms. We clarified that in the text (page 9, line 9).

Reviewer #2 (Remarks to the Author):

This manuscript addresses an important question in cryo-ET data collection - what's the best data collection scheme for subtomogram averaging. The insight provided by the systematic analysis will be of great interest to the growing cryo-ET user community. Considering its potential high impact, the current version needs clarification and updates.

(1) Methods-Image Acquisition says only continuous scheme was collected using the built-in function in SerialEM. SerialEM has built-in function for bidirectional too (TS setup-Run series in two directions). Why the authors chose not to use SerialEM's built-in function? Please compare the throughput of the built-in function with the customized script.

As pointed out in the general note above, the major objective of our paper is to assess how the distribution of dose affects the attainable resolution in subtomogram averaging. We chose the given implementation for the following reasons:

Hagen et al. 2016 describes several measures to overcome hardware limitations that to this date were not fully implemented in the SerialEM tilt-controller (for any tilt-scheme). As we did not want any (possibly small) differences in the final outcome between the bidirectional and the dose-symmetric scheme to be contributed to the differences in the implementation a script was used for the bidirectional and dose-symmetric tilt-schemes. This allowed us to keep stage movements similar, especially the tilt backlash on the negative tilt branch which is not possible in the SerialEM tilt-controller. In the continuous tilt-scheme the tilt direction does not reverse and thus the tilt-controller implementation was sufficient for a fair comparison. We used 10 s wait-after-tilt delay time which corresponds to typical drift measurement times of the other tilt-schemes.

As correctly pointed out by the reviewer in question #3, there is a beta version of SerialEM with integrated dose-symmetric scheme in tilt-controller which enables to compare the three schemes using the same less conservative approach. However, the dose-symmetric implementation is still undergoing development and is currently not suitable for benchmarking purposes. While it would indeed be interesting to measure the impact of the conservative script approach in comparison to the SerialEM tilt controller implementation, such comparison would require systematic testing of all tilt-controller parameters which is in practice not feasible.

As for the throughput of the tilt-controller version of the bidirectional scheme – it takes on average 20 minutes to collect 41 images.

(2) Besides the schemes tested in this manuscript, another modified bidirectional scheme is worth investigating. In this scheme, the first tilt starts at intermediate tilt angles, say 30 deg. From +30 deg, it tilts toward 0, passing 0 and continuing toward negative angles until reaching -60deg. From +30 to -60, an image is recorded every 3deg (31 tilts in this first part). After that, it goes back to 30 deg, continuing toward +60deg and taking one image every 3deg (10 tilts in this second part). By doing this, the relatively less damaged data were collected at relatively low tilt angles. It is a compromise between throughput and retaining the high resolution information at low angles. I suggest the authors add this to their comparison. One complication is that all the existing datasets are collected on one single grid, and it may not be possible to use the same one for this additional test. Nevertheless, the results can enhance the completeness of the manuscript.

This is a very valid point and has been raised by both reviewers. We have acquired additional dataset on a different grid using bidirectional scheme with [-20:61] followed by [-23:-59], together with a dose-symmetric control dataset on the same grid such that at least these two new datasets are comparable. In both cases (on both grids), the resolution obtained with the two experiments with dose symmetric schemes was almost the same and the resolution obtained in the two experiments with the different bidirectional schemes (with and without offset) was similarly lower. Please see also our response to the first reviewer and Supplementary, Section 1 for more details.

(3) Recently, SerialEM's development version incorporates built-in functions for dose-symmetric as well. More users probably would prefer to use the built-in function. The authors should benchmark the built-in DS scheme using parameters as close to their native Hagen scheme as possible and report if the built-in DS scheme gives a higher throughput or other advantages. Update Table 2 accordingly. Ref: <http://bio3d.colorado.edu/ftp/SerialEM/History-beta.txt>

Again, as pointed out in the general note above, the major objective of our paper is to assess how the distribution of dose affects the attainable resolution in subtomogram averaging. We included a table with the performance into the manuscript for information, but we do not have the capacity to systematically benchmark different acquisition software. Our motivation was the following:

The built-in dose-symmetric scheme is still under development and until very recently lacked functionality to get even close to the scripted version we use. For now, it is reasonable to assume that the relative difference in the outcome between the dose-symmetric and bidirectional scheme would remain the same for their tilt-controller versions. Timewise, using as conservative settings as possible, the acquisition time using the tilt-controller is similar to the bidirectional tilt-controller scheme (with conservative stage-tilt delay of 10s). This might however change with further development.

(4) Methods say 20~30 tilt series per scheme were collected, but only 8~10 tilt series were used for further processing. Is it common practice in the authors' lab to throw away >50% of raw data at this stage? Please comment on this practice in the manuscript, to give the audience an idea about the importance/necessity of getting rid of poor data early on.

This strict sub-selection was done in this particular case in order to eliminate potential quality differences, i.e. in specimen thickness even on the same grid, as much as possible. Obviously, this cannot be considered common practice because it will not be possible for most samples in which the number of asymmetric units is limiting. We had pointed this out on the page 10, line 30. The remaining data are not necessarily poor, but yielded slightly lower resolution.

(5) Related to (4), page 11 says 8% of picked particles were left. Please provide more details on the criteria used for the clean up (quantitatively if possible). How sensitive is the final result to this step? Will slightly more/less strict criteria make a big difference?

We added more detailed description of our cleaning procedures (see Supplementary section 2), including the numbers of particles before and after the cleaning (see the Table S3). The 8% refers to the initial positions that 10x oversample the actual lattice (as mentioned in the section on Particle picking) and thus correspond to roughly 80% of actual subunits – we clarified that in the manuscript (page 11, line 32). As shown in the Table S3, the majority of particles is discarded based on distance corresponding to their

physical distance within the lattice. In the work by Schur et al. 2016, the misaligned particles were cleaned manually based on a cross-correlation (CC) threshold which had to be found for each VLP. To avoid a user bias we replaced this step by the geometric cleaning which is fully automatic and fully based on data quality, i.e. it does not require any user-specified parameters. As shown in the Supplementary Figure 4, this approach works for both full and incomplete VLPs.

Using slightly more/less strict criteria for the geometric cleaning would lead to only minor changes in the number of particles but would not affect the final outcome as only a random subset of particles is used for the final subtomogram averaging (see Table S3).

In general, the B-factor analysis provides a likely estimate of the cleaning impact - with strict cleaning most likely slightly outperforming the resolution in Figure 2 (panel d,e).

(6) Among the software used (Table S1), some are common, such as eTomo and CTFFind, while others are not, such as Dose Exposure correction (matlab scripts), and novaCTF. Please comment in the manuscript on the importance of using these less common tools and the computational cost, whether they should be used routinely for all subtomogram projects, or used selectively for specific projects. I understand each of them has its own publication, but it would be helpful to have a few comments for the audience to grasp the (critical) roles they play in the process and whether they should be widely adopted in the cryo-ET community.

Again, it was our major objective to assess how the distribution of dose affects the attainable resolution in subtomogram averaging. We used the processing pipeline that is implemented in our laboratory in a systematic manner to justify the respective conclusions. We will not be able to compare these tools to alternative software solutions that we do not have implemented. As the reviewer mentions, it has been previously shown even on this particular sample, that dose exposure correction (Schur et al. 2016) and 3D-CTF correction (Turanova et al. 2017), is beneficial for improving the resolution. Both methods are implemented also in subtomogram packages such as emClarity or Warp and can also be found in beta version of eTomo. In other words, they are already being more routinely incorporated to SA workflows.

(7) Table S2 says a 10s delay is used. Is it really necessary? For 41 tilts, that's almost 7min. Is 10s an empirical number or the authors have done some tests and found 10s gives the optimal results?

The delay of 10s corresponds to the typical time needed for drift stabilization during dose-symmetric acquisition. As mentioned in our response to the question #1, we used the same delay for the continuous scheme to make sure we were collecting as close to the dose-symmetric scheme conditions as possible. We clarified that in the legend of Table S2.

(8) Table S2 says Beam intensity is kept constant, is exposure also kept constant or varied. I suggest the authors replace Table S2 with a screenshot from SerialEM Tilt series Setup, so that it's easy to follow which parameters are used.

The exposure time is also kept constant throughout (we added this on page 10, line 7). Since the SerialEM software continues to be developed the screen shot would have a very short half live of usefulness. We thus would prefer to keep the table.

(9) Table 1 shows a defocus gradient is used. How is this achieved? SerialEM's TS setup only allows one defocus value to be specified. When SerialEM's built-in TS function is used, did the author use a script or just manually change the TS setup every few series? Please comment on this in the manuscript. My personal experience is that even when a script is used in the initial-task to vary the target defocus, SerialEM only takes the value from the TS setup. This seems to be a hurdle for automated data collection.

We set the defocus manually per tilt-series in the SerialEM Navigator Tilt Series Parameters option. We added this information to the legend of the Table S2.

(10) Methods-Acquisition. Please specify the type of holder, the dual axis holder typically on the Krios G2, or the single axis holder on G3? I'm wondering if the (presumably more stable) single axis holder alleviates the problem of losing field of view.

We have used our Krios G1 serial number 3208, meaning system #8 with a dual-axis holder (we added this information on page 10, line 3).

High-resolution sub-tomogram averaging requires working at high magnifications, hence it also requires full tracking for each tilt to alleviate the problem of losing field of view. Our experience with the single axis holder on our Krios G3i serial number 3686, meaning system #86, is that drift stabilizes faster. The shifts on this single axis system, which was the first single axis holder system ever shipped, are much higher than on our dual axis system.

(11) Methods-Acquisition. Please specify the version of SerialEM, as specific as possible. SerialEM undergoes constant changes. Version number is necessary for reproducibility purposes. Also add the version of GMS (DM). These information may be incorporated into Table S1.

We added the information to the Table S1.

(12) Have the authors attempted to increase the throughput utilizing the FrameSeriesFromVar function in SerialEM? This function allows the user to potentially skip focusing and/or tracking steps by feeding a list of calibrated focus/image shift values to the program. FEI tomography offers this option for years. Ref: http://bio3d.colorado.edu/SerialEM/betaHlp/html/script_commands.htm

As mentioned above, our objective was not to optimize throughout. In any case, this method does not work at the magnification used in our study (see Eisenstein et al. 2019 - <https://doi.org/10.1016/j.jsb.2019.08.006>).

(13) For in-focus VPP dataset, did the authors go the extra mile to improve the accuracy of focusing, such as iterating the defocus measurement at opposite sides? Table 2 shows 'DS VPP foc' takes the longest time.

No, but we iteratively use AutoFocus to obtain less than 0.1 um change in focus (as described in Schur et al. 2013, <https://doi.org/10.1016/j.jsb.2013.10.015>). As the estimation is more difficult for the data in focus this step prolongs the acquisition times.

(14) Page7, L7: “The visual inspection of the respective structures does not credibly support the estimated resolution (Fig 4)”. Please add the nominal resolution as insets to each panel in Fig. 4. Add a panel of the density map from bi-directional scheme to Fig4. Line2 says the FSC shows VPP result is similar to bi-directional. It is helpful to see a comparison of the density maps of the two schemes.

We added the insets with the nominal resolution to each panel in Fig 4. The sentence on line 2 erroneously compared the VPP datasets to the bidirectional one – this comparison was thus removed. As shown in Figure 2, the bidirectional scheme outperforms both VPP datasets. Consequently, we did not add the map from the bidirectional scheme into Figure 4 (at this resolution it would be very similar to the dose-symmetric one).

(15) Methods-Sample preparation. What does ‘degassed’ mean? How is it performed? Add the type of instrument used for glow discharge. Add the supplier of the colloid gold. Add the type of blotting paper.

Term ‘degassed’ means that the grids were stored in a vacuum desiccator. We clarified that in the manuscript as well as added all requested information (see Sample preparation paragraph on page 9).

(16) Page 10, particle picking. Add the initial number of particles.

We added a table with number of VLPs as well as starting subtomograms in Supplementary, Table 3. However, we would like to stress here that the initial number of subtomograms is not directly relevant for the final structure as we heavily oversample the VLP lattices (as described in the Particle picking paragraph on page 11).

(17) Page 8, second paragraph essentially says VPP is not a good idea for subtomogram averaging aiming for high resolution, given the tools available now. Other cryo-ET scientists have given similar advice for years. Now we finally have published results to support it. I feel the authors can strengthen the conclusion, giving the audience a clearer message.

We rephrased our conclusions on VPP and clearly stated that it is not suitable for high-resolution SA (page 9, line 4).

Terminology issues:

1) Minimum tilt angle (e.g. page 3, L6). I believe it refers to the most negative tilt angle, which is actually a high tilt angle. Calling it minimum is ambiguous. Change it to the most negative tilt angle or something less ambiguous.

2) The lowest defocus (page 10, L7). Based on the context, I believe it refers to the most underfocus value. Defocus means out of focus, which can be over- or under-focus. When I hear low defocus, I relate it to close to focus. Change it to the highest underfocus or something less ambiguous.

Thank you for your suggestions. We changed the “minimum tilt angle” to “minimum angle” and “the most underfocused” throughout the text (where appropriate).

Minor correction:

Page 2, line 8: The most important properties are ...

Page 12, line 5: For each dataset, 3 logarithmically smaller ...

Thank you for spotting this – we corrected both.

Reviewers' Comments:

Reviewer #1:

Remarks to the Author:

In my opinion, the authors have addressed the reviewers concerns. My only further request is that the local alignment analysis provided in the rebuttal be provided to the reader in the Supplementary Figures/Information. Even though the result is practically null, this analysis is instructive for readers who are considering the slew of potential collection and analysis methods in cryoET, which is the point of the manuscript.

Reviewer #2:

Remarks to the Author:

REMARKS TO THE AUTHOR

The authors have sufficiently addressed my questions. I acknowledge the authors' argument on using scripts to keep stage movement similar. The current version is suitable for publication.

There is a typo in Fig.4, two panels are labeled 'b'. The second row should be 'c'.

We want to thank the reviewers for their positive remarks.

Reviewer #1 (Remarks to the Author):

In my opinion, the authors have addressed the reviewers concerns. My only further request is that the local alignment analysis provided in the rebuttal be provided to the reader in the Supplementary Figures/Information. Even though the result is practically null, this analysis is instructive for readers who are considering the slew of potential collection and analysis methods in cryoET, which is the point of the manuscript.

We added the local alignment analysis as a separate section into the Supplementary.

Reviewer #2 (Remarks to the Author):

The authors have sufficiently addressed my questions. I acknowledge the authors' argument on using scripts to keep stage movement similar. The current version is suitable for publication. There is a typo in Fig.4, two panels are labeled 'b'. The second row should be 'c'.

Thank you for spotting this – we corrected it.